# Study on the Influence of Unbalanced Phase Difference Combinations on Vibration Characteristics of Rotor Systems

**DOI:** 10.3390/s25061691

**Published:** 2025-03-08

**Authors:** Yiming Cao, Shijie Zhong, Xuejun Li, Mingfeng Li, Jie Bian

**Affiliations:** 1Key Laboratory of Rotor Vibration Monitoring and Diagnosis Technology for Machinery Industry, Foshan University, Foshan 528000, China; caoyiming@fosu.edu.cn (Y.C.); zsjie5009@163.com (S.Z.); 2Aecc South Industry Company Limited, Zhuzhou 412000, China; lmf18007330425@163.com; 3AVIC Hunan Power Machinery Research Institute, Zhuzhou 412002, China; bianjie_hrbeu@163.com

**Keywords:** rotor dynamics modeling, unbalanced phase, eddy current sensor, rotor dynamic balance, vibration response

## Abstract

Taking the cantilever rotor of a turbine engine as the research object, a dynamic and finite-element model of the cantilever rotor is established, and the effectiveness of the model is verified by the rotor test platform. The transfer function method is used to balance the rotor system under unbalanced excitation, and the experiments prove that the method adopted in this paper has a good balancing effect and effectively reduces the vibration of the unbalanced rotor. On this basis, the experimental tests and simulation analyses of the rotor vibration response under different unbalanced phases and difference combinations are carried out, and the influence of the unbalanced phase’s difference combinations on unbalance and dynamic balance is analyzed. The results show that the vibration response of the system decreases with the increase in the unbalanced phase difference combinations, and the amplitude of the vibration induced by the unbalance of the reverse combination is smaller than that of the in-phase combination. The work in this paper can provide a theoretical basis for the dynamic balance and vibration control of the flexible rotor of an aero-engine.

## 1. Introduction

Turbine engines operate in harsh service environments, typically under conditions of high temperature, high pressure, and high rotational speed. The structural configuration of the rotor system in turbine engines is intricate, with its power turbine output shaft being a slender shaft that terminates in a two-stage free turbine. During normal operation, air enters the engine through the intake, is pressurized by the compressor, and is then further pressurized by the centrifugal turbine before reaching the combustion chamber. Within the combustion chamber, the air thoroughly mixes with fuel and burns, generating high-temperature, high-pressure gases that are ejected towards the outlet at high speeds. This process harnesses the immense energy needed to drive the rotation of the power turbine. As the core component for energy conversion, the power turbine transmits torque through the slender shaft to the front end of the engine. Notably, this slender shaft not only needs to traverse the combustion chamber and the compressor rotor cavity but must also maintain sufficient strength and stability to ensure reliable torque transmission. Therefore, in turbine engine design, the power turbine rotor is typically configured as a slender, hollow flexible rotor, with the overhung rotor structure being its quintessential configuration [1,2]. The distinguishing feature of the overhung rotor structure is that the impeller or wheel disk is usually suspended from one end of the support bearing. However, due to manufacturing and installation errors, as well as wear and tear over extended operation, unbalanced masses often concentrate at this end. The slender shaft structure at the other end will inevitably produce strong unbalanced vibration under extreme conditions such as high speed, high temperature, heavy load, and supercritical conditions. These unbalanced vibrations not only affect the equilibrium state of both ends of the rotor support but also complicate the vibration response characteristics of the rotor system. Consequently, in the design and optimization process of turboshaft engines, these factors must be thoroughly considered to ensure the stable operation and reliability of the rotor system.

Rotor imbalance is a primary excitation source in rotating machinery. Excessive imbalance will lead to rotor deformation and internal stress, intensifying vibration and noise, and even triggering safety accidents. Therefore, the influence of imbalance on rotor dynamic characteristics has been a focus of attention for scholars both domestically and internationally [3,4,5,6,7,8,9,10,11,12]. Regarding rotor imbalance issues, Kang et al. [13] investigated counter-rotating dual rotor systems and discovered counter-rotating phenomena induced by imbalance excitation, revealing that significant imbalance in one rotor is a crucial factor causing counter-rotation of the other rotor. Song et al. [14] established a bilateral imbalance prediction model for multi-stage assembled rotors to enhance initial assembly qualification rates, thereby accurately predicting bilateral imbalance and its phase effects in multi-stage rotors. Xiang et al. [15] analyzed the impact of residual unbalanced mass on vibration amplitudes of magnetically suspended flywheel rotors by measuring dynamic displacements. Nayek et al. [16] proposed an online imbalance identification method based on modeling and simulation to determine the eccentricity and direction of the spindle, addressing dynamic imbalance issues caused by uneven mass distribution in rotors. Jiang et al. [17,18] introduced a method for unbalanced compensation in rotor systems by generating control signals based on the real-time position of rotor unbalanced mass. Han et al. [19,20,21] found that the axial position of rotor imbalance has a significant impact on high-speed dynamic balancing and, leveraging deep learning, they proposed a flexible rotor imbalance position identification method which was based on dynamic model simulations and experimental data.

However, the aforementioned extensive research primarily focuses on the impact of imbalance magnitude and position on the dynamic characteristics of rotor systems, with less consideration given to the influence of imbalance phase, especially in terms of combinations of imbalance phases for cantilever rotor types. Ma et al. [22] analyzed the effects of imbalance position, imbalance magnitude, imbalance phase, and bearing stiffness on the vibration response of dual rotor systems (both inner and outer rotors). Wang et al. [23] conducted parameter identification of bearing stiffness and damping coefficients for single-span rotor systems, considering the functional relationship between imbalance response and position, rotor imbalance, bearing stiffness, and damping coefficients. Han [24], Wang [25], and Spagnol [26] investigated the coupling effects of imbalance angle and magnitude on crack breathing behavior, discovering that imbalance can affect the crack breathing mechanism, which in turn influences the dynamic characteristics of cracked rotor systems. Zhou et al. [27] proposed an online imbalance compensation algorithm for active magnetic bearings based on the least mean squares method and the influence coefficient method, considering the effects of different imbalance masses and phases in rotors. Research considering the imbalance phase has played a significant role in guiding the positioning of installed balancing masses [28], highlighting the necessity to study the influence patterns of combinations of imbalance phases on the dynamic balancing of rotor systems. Wang et al. [29] established a finite element model of the rotor system and found that merely changing the phase combination of residual imbalance can significantly reduce the vibration levels of rotor systems. Wang et al. [30] discovered that imbalance phase is a key parameter affecting rotor vibration characteristics, with certain combinations of imbalance phases tending to induce energy transfer between shafts and bearings. Chen et al. [31] analyzed the phase difference between two unbalanced rotors and the motion lag phase of the rigid frame and studied the coupled motion characteristics of unbalanced rotors based on this. However, the aforementioned research lacks a more detailed and comprehensive analysis of the impact of combinations of rotor imbalance phase differences on vibration characteristics, particularly regarding the effects of these combinations on rotor dynamic balancing effectiveness.

Vibrations induced by imbalance will accelerate the wear of components such as bearings and shaft seals, thereby shortening the service life of equipment. Through dynamic balance correction, the wear of these components can be reduced, thereby extending the overall lifespan of the equipment [32]. Chung et al. [33] proposed an automatic rotor balance detection method based on adaptive vision, enabling the rapid and efficient dynamic balancing of rotor systems. Wang et al. [34] introduced a dynamic balance optimization model for spindle rotor systems based on the particle swarm algorithm. Zhang et al. [35] utilized a transfer function to solve the relationship between imbalance and vibration response, proposing a rotor dynamic balance optimization method based on the Grey Wolf Optimizer (GWO). Bin et al. [36] presented a virtual dynamic balance method for multi-rotor systems that does not require trial weights. Gyan Ranjan et al. [37] developed an Advanced Influence Coefficient Method (AICM), which makes use of influence coefficients obtained at high speeds and imbalances that are identified at low speeds to effectively estimate the balancing masses required for high-speed flexible rotor balancing. Yao et al. [38] studied a dynamic balancing method for multi-speed flexible rotors combined with a Dual Objective Method (DOM), overcoming the issue of excessive residual vibrations produced by the Least Squares Method (LSM) when performing dynamic balancing of flexible rotors at specific rotor speeds. Wang et al. [39] further explored the influence of rotational speed and rotor imbalance on deviations in the static and dynamic balance positions of rotors. They found that due to the action of the rotational imbalance forces, there is a significant deviation between the dynamic balance position and the static balance position of the rotor, and this deviation has an approximately linear relationship with rotational speed. Zhao et al. [40] considered the total unbalance and local maximum unbalance of the combined rotor and proposed a dual-objective optimization method to reduce the unbalance, which improved the accuracy of predicting the eccentricity value and eccentricity phase.

However, an analysis of the aforementioned literature reveals that the previous literature has certain limitations in the research on the unbalanced phase difference in rotors. For example, the number of phase combinations is relatively small, which is insufficient to draw regular conclusions. In this paper, not only are seven combinations of unbalanced phase differences considered, but factors such as the magnitude and position of the unbalanced mass are also comprehensively taken into account. By establishing a comprehensive dynamic model and conducting a large number of experimental verifications, the vibration response laws of the rotor system under the coupling effect are deeply explored. The influences of the different combinations of phase differences (such as reverse combinations, in-phase combinations, etc.) on the vibration response and dynamic balancing effect of the rotor system under different working conditions are studied, and the internal relationship and variation laws between the combinations of phase differences and the vibration response are revealed. The sections of this paper are organized as follows: Section 2 describes the dynamic modeling approach for turbine rotor systems. Additionally, natural characteristic experiments are conducted based on the established rotor test bench to validate the effectiveness of the constructed model. Section 3 performs dynamic balancing on the turbine rotor system using the transfer function method and verifies the accuracy of the balancing results. Based on this foundation, the influence patterns of the combinations of imbalance phase differences on the vibration response and dynamic balance results of rotor systems are investigated. Some main conclusions are presented in Section 4.

## 2. Rotor Dynamics Modeling Method

### 2.1. Turbine Rotor System

The power turbine rotor system of a certain aero-engine studied in this paper is illustrated in Figure 1. It primarily consists of a slender shaft, a short shaft, support bearings, and turbine disks. The fixed bearing on the left side of the rotor system supports the slender shaft, while the cylindrical roller bearing on the right side supports the short shaft. The short shaft and the slender shaft are connected through an interference fit and transmit torque via splines. The short shaft and the turbine disks are located to the right of the slender shaft. The first-stage turbine disk mates with the second-stage turbine disk through a rabbet and end face, and they are securely fastened together with bolts. The stub shaft and the second-stage turbine disk are also fastened together with bolts. The two-stage turbine disk can be equated to a rigid rotary disk.

### 2.2. Dynamic Modeling of Flexible Cantilever Rotor System with Elastic Support

To establish a dynamic model for the power turbine rotor, simplifications of the rotor are necessary. In this paper, the following assumptions are made: (1) the hollow slender shaft is equivalent to a solid shaft with an equal bending stiffness; (2) the two-stage turbine disks are simplified as lumped mass disks; and (3) the bearing supports are considered as elastic supports. Based on these assumptions, the turbine rotor system is discretized into a series of rotor shaft elements, disk elements, and elastic support stiffness elements. Consequently, the differential equations of motion for the rotor system can be derived as follows:(1)$$M\ddot{q}+(C+\omega G)\dot{q}+Kq=F$$
where ***q*** is the displacement vector of the rotor system, and *ω* is the rotor speed. ***M*** is the mass matrix of the rotor system. ***K*** is the stiffness matrix of the rotor system. ***G*** is the gyro matrix of the rotor system. ***F*** is the rotor unbalance force vector. ***C*** is the damping matrix of the rotor system. The system damping is calculated in the form of Rayleigh damping through the stiffness matrix of the system:(2)$$C=\alpha_{0}M+\beta_{0}K$$
where the expressions of *α*_0_ and *β*_0_ are shown in Ref. [41].

The unbalanced force vector at node i can be expressed in the following form:(3)$$F^{i}={\left[\begin{array}{cccccc} F_{ui} & F_{vi} & 0 & M_{ui} & M_{vi} & 0 \end{array}\right]}^{T}$$
where *F_ui_*, *F_vi_*, *M_ui_*, and *M_vi_* are the rotor unbalance forces and moments located at node *i*, which can be expressed as follows:(4)$$\left\{\begin{array}{c} \begin{array}{l} F_{ui}\\ F_{vi}\\ M_{ui}\\ M_{vi} \end{array} \end{array}\right\}=m_{i}e_{1i}\omega^{2}\left\{\begin{array}{c} \begin{array}{l} \cos(\omega t+\varphi_{i})\\ \sin(\omega t+\varphi_{i}) \end{array}\\ \begin{array}{l} -e_{2i}\sin(\omega t+\varphi_{i})\\ e_{2i}\cos(\omega t+\varphi_{i}) \end{array} \end{array}\right\}$$
where $\varphi_{i}$ is the initial phase of the unbalance on the blade disk. *m_i_* is the blade disk mass. *e*_1*i*_ and *e*_2*i*_ are the radial and axial eccentricity, respectively. *i* indicates the location of the blade disk. Therefore, the vibration response of the rotor system is not only related to the unbalance amplitude and position, but also to the phase of the unbalance. The effect of the unbalanced phase combination on the vibration response of the system is studied. If $\varphi_{i}=\varphi_{j}$, the unbalance of the two blade disks will form an in-phase combination unbalance. If $\varphi_{i}-\varphi_{j}=\pm 180^{{}^{\circ}}$, the unbalance of the two blade disks forms an inverse phase combination unbalance.

After neglecting the influences of factors such as bearing clearance and oil film thickness, the rotor support system simplifies the bearing as a spring-damper element acting at the rotor pivot point. Assuming that the coupling node between the support bearing and the shaft segment is denoted as *i*, as shown in Figure 1, the support stiffness of the bearing is *K*. In the interference fit contact between the slender shaft and the short shaft, the connection between the contact pairs is regarded as a spring-damper element capable of transmitting forces and displacements. Assuming that the *i*th node on the slender shaft and the *j*th node on the short shaft are in interference fit, the spring-damper element is connected at one end to the *i*th node of the slender shaft and at the other end to the *j*th node of the short shaft, as shown in Figure 1, with a contact stiffness of *K*. Hence, as follows:(5)$$C_{b}{\dot{q}}_{i}+K_{b}q_{i}=F_{b}$$
where ***C****_b_* is the supporting damping matrix, ***K****_b_* is the supporting stiffness matrix, ***F****_b_* is the external excitation vector, and ***q****_i_* is the displacement vector of the element node of the rotor system.

The unbalanced vibration response ***q*** of the rotor system is the linear superposition of various modes under unbalanced excitation:(6)$$q=\sum_{n=1}^{\infty }\left[q_{cn}\cos(n\omega t)+q_{sn}\sin(n\omega t)\right]$$
where ***q****_cn_* is the real part of the nth mode response ***q****_n_*, ***q****_sn_* is the imaginary part of the *n*th mode response ***q****_n_*. The finite element model of the rotor system finally established is shown in Figure 2, and the relevant parameters are shown in Table 1.

### 2.3. Model Verification

To validate the effectiveness of the established finite element model of the rotor, this paper designs and constructs a turbine rotor system testbench, as shown in Figure 3. The rotor testbench employs a high-performance electric spindle motor as the driving component, with a motor power of 15 kW and a rated voltage of 380 V; the rotor system is supported in a three-point configuration. The two turbine disks, which are suspended on the right side of support 3, feature threaded holes for setting an initial imbalance. There are four bosses on the shaft that can serve as counterweight planes, equipped with weight rings for adding or subtracting mass. To accurately measure the vibration displacement signals of the rotor system, four high-sensitivity eddy current displacement sensors (type: RP6600XL (Kaihang, Huanggang, China), precision: <0.1%) are installed, ensuring data accuracy and reliability. The counterweight ring is a two-half ring structure with 6 through holes for easy application of counterweight mass. During the experiments, to verify the accuracy of the finite element model, modal analysis using the hammer impact method is conducted to validate its natural characteristics.

A virtual prototyping modeling of the rotor system studied in this paper is conducted through simulation, as shown in Figure 4a. Among them, supports 1 and 2 can be approximated as rigid supports, while support 3 is an elastic support with stiffness settings according to the data in Table 1. A preset initial unbalanced mass of 20 g is simulated on the turbine disk with a phase of 0° (with the Y-Z plane as the rotation plane, defining the phase of the +Z axis as 0). After completing the settings, the model is meshed, as shown in Figure 4a, where the disk and the shaft section at support 2 are tetrahedral elements, and the other parts are hexahedral elements. By solving the model, the first-order critical speed of the rotor system is obtained as 6263.1 rpm (104.4 Hz) and the second-order critical speed as 15,729 rpm (262.2 Hz). Based on the vibration mode shapes of the rotor system, dynamic balancing measurement points are set, as shown in Figure 4b. The first-order vibration mode is the U-shaped bending in the middle of the shaft near the disk, and the second-order vibration mode is also U-shaped bending. The measurement points are set at the anti-nodes of the vibration mode shapes, with measurement points 1 and 2 at the middle two shaft sections of the entire transmission shaft, measurement point 3 near the middle of the shaft close to support 3, and measurement point 4 at the turbine disk. An acceleration sensor (type: PCB-356A01 (PCB, Buffalo, NY, USA)) is installed at support 3 of the testbench, and the frequency response function curve obtained through the hammer impact method, as shown in Figure 5, indicated that the first-order natural frequency of the rotor system is 111.3 Hz and the second-order natural frequency is 281.4 Hz. After calculation, the error between the experimental and simulated first-order natural frequencies is 6.2%, and the error between the experimental and simulated second-order natural frequencies is 7.3%; the reason for the error may be due to the simplification of the model. Sensor accuracy has been caused by assembly errors. The error in the control range verifies the validity of the model. This finding not only provides a solid theoretical foundation for subsequent rotor dynamics research but also offers powerful technical support for rotor system design in practical engineering applications.

## 3. Analysis of Unbalance Phase Combined Vibration Response of Cantilever Rotor

### 3.1. Dynamic Balance of Rotor System Based on Transfer Function Method

#### 3.1.1. Simulation Calculation

In order to study the law of the influence of unbalanced phase combinations on the vibration balance of the rotating subsystem, the transfer function method [35] is adopted in this paper to solve the balance required by the rotor system. The advantage of the transfer function method is to replace the influence coefficient with the transfer function value, so that there is no need for multiple start—stop operations, which can cause a waste of resources and economic costs. The rotor system studied in this paper is equipped with 4 measuring points and 4 counterweight planes. According to the actual working range of the test bench, the rotor dynamic balance is carried out at speeds of 4860 r/min (81 Hz) and 5400 r/min (90 Hz) based on the transfer function method. According to the established finite element model, the transfer function ***H***(*Ω*) is defined by the ratio of the output vibration response to the input excitation, and the transfer function matrix at a certain speed can be written as follows:(7)$$H\left(\Omega \right)=\left[\begin{array}{cccc} H_{11} & H_{12} & H_{13} & H_{14}\\ H_{21} & H_{22} & H_{23} & H_{24}\\ H_{31} & H_{32} & H_{33} & H_{34}\\ H_{41} & H_{42} & H_{43} & H_{44} \end{array}\right]$$
where the subscript *i* in *H_ij_* indicates the measuring point *i*, and the subscript *j* indicates the input excitation on the counterweight plane *j* (excitation is 10 g, 0°). When the speed is 5400 r/min, the frequency response curves of the four measuring points are shown in Figure 6.

The relationship between the influence coefficient matrix and the transfer function matrix is as follows:(8)$$\alpha \left(\Omega \right)=\Omega^{2}H\left(\Omega \right)$$
where ***α***(*Ω*) represents the influence coefficient matrix, ***H***(*Ω*) represents the transfer function matrix, and *Ω* represents the rotor speed. The influence coefficient method obtains each column element of the influence coefficient matrix by adding test weights in the counterweight plane one by one. The transfer function rule is used to measure each column element of the transfer function matrix by hammering method or harmonic response analysis. The weight and phase of each counterweight plane of the rotor system under the unbalanced initial phase are obtained by using the linear relationship between the correction amount and the measured amount in the influence coefficient method, as shown in Table 2.

#### 3.1.2. Experimental Verification

In order to verify the accuracy of the dynamic balance results, the vibration test is carried out on the test bench. The test speed is 5400 r/min. Firstly, the vibration response of four measurement points of the rotor system is measured and the vibration displacement is extracted in the original state before dynamic balance. According to the calculation results in Table 2, the corresponding unbalance mass (bolt and nut) and phase are artificially added at the counterweight planes 1, 2, 3, and 4, respectively. The experiment is conducted again in the equilibrium state after dynamic balance, and the vibration displacement in this state is extracted. The comparisons of rotor system vibration response measured at point 1 and point 3 before and after trim are shown in Figure 7. The results show that the dynamic balancing scheme based on the transfer function method can suppress the unbalanced vibration of the rotor system well. Through dynamic balancing, the amplitude of the rotor system at two measuring points can be reduced by about 50% at 5400 r/min.

### 3.2. Effect of Rotor Phase Difference Combination on Unbalance Vibration Response

It is easy to produce random combination unbalance among the components of the rotor, resulting in complex vibration response characteristics of the rotor. At present, the influence of unbalance magnitude and position on rotor vibration response has been studied, but there are few studies on unbalance phase difference combinations. In order to study the influence law of unbalanced phase difference combination on vibration response of rotor system, nine holes are evenly divided in the range of 0–180°, and seven groups of phase difference combinations are selected, respectively, to set unbalance, as shown in Figure 8. In this section, the effects of the unbalanced phase difference combination on the vibration response of the rotor system are studied from the perspectives of simulation and experiment.

#### 3.2.1. Simulation Analysis

To study the influence law of unbalanced phase difference combinations on the vibration response of the rotor systems, seven sets of phase difference combination are selected between 0 and 180° (0°, 22.5°, 67.5°, 90°, 112.5°, 157.5°, and 180°), and harmonic response analysis is performed on them, respectively; the amplitude diagram of the measuring point 4 are obtained as shown in Figure 9. It can be seen from the figure that the vibration amplitude of the system caused by rotor unbalance decreases with the increase in the unbalance phase difference. At first order critical speed, the amplitude of the vibration induced by the unbalance of the reverse phase combination is reduced by 72% compared with that of the unbalance of the in-phase combination. The amplitude of the vibration induced by the unbalance of the reverse phase combination at the second-order critical speed is 59% of that of the unbalance induced by the in-phase combination.

#### 3.2.2. Test Verification

To verify the effect of unbalanced phase difference combination on the vibration response of rotor system, the rotor unbalanced vibration experiment is designed and compared with the simulation results. The test bench built in Section 2.3 is used to conduct the rotor unbalanced vibration response experiment, as shown in Figure 10. The speed is set to 5400 r/min, and the vibration response data collected by the sensor at measurement point 1 are extracted under seven sets of unbalanced phase difference combinations, as shown in Figure 11a. Compared the vibration data of different phase combinations at *t* = 0.04 s, as shown in Figure 11b, it can be seen that the law of the experimental data are consistent with that of the simulation. The vibration response of the rotor system decreases with the increase in the phase difference, and the amplitude of vibration induced by the unbalance of the reversed phase combination is smaller than that of the unbalance of the in-phase combination, which verifies the accuracy of the simulation results. At the same time, the effect of unbalance phase difference cannot be ignored in rotor unbalance problem.

#### 3.2.3. Analysis of Dynamic Balance Under Different Phase Difference Combinations

The accuracy of the simulation results has been verified in the previous section. Based on this, this section will continue to analyze the influence law of the unbalanced phase difference combination on the rotor dynamic balance results. When the rotation speed is 4860 r/min and 5400 r/min, the balance quantities at counterweight planes 1, 2, 3, and 4 under the unbalanced phase difference combinations of the seven groups are calculated, respectively, as shown in Table 3 and Table 4. In Figure 12, the results of dynamic balance under two rotational speeds show that the absolute values of the balance mass of the four counterweight planes under the phase difference combination of 90° are smaller.

## 4. Conclusions

(1)The rotor dynamic model based on the finite element method is analyzed by the transfer function method, and the vibration test bench is set up to verify the validity of the simulation results. On this basis, seven kinds of unbalance phase difference combinations are defined, and the influence of unbalance phase difference on rotor unbalance response and dynamic balance results is studied.(2)In the range of the first critical speed, the vibration amplitude caused by the unbalance of the reversed phase combination is reduced by 72% compared with that of the unbalance of the in-phase combination, and the vibration amplitude caused by the unbalance of the reversed phase combination is 59% of that of the second-critical speed. For the dynamic balance results, when the phase difference is 90°, the absolute value of the balance mass of the four counterweight planes is small. The combined vibration response law of the cantilever turbine rotor of the turboshaft engine with unbalanced phase difference considering the dynamic characteristics of the elastic support is studied in this paper, which can provide reference for rotor vibration fault analysis. It is worth noting that the conclusions of this paper are obtained in the research context of a cantilever rotor system. Due to the differences in factors such as mass distribution, stiffness distribution, and support conditions among different rotor systems, in practical applications, the conclusions of this study cannot be directly applied to other rotor systems. Instead, it is necessary to conduct analyses and experiments based on the specific rotor structures and parameters. Moreover, in practical engineering applications, the impacts of more complex factors need to be considered, such as the nonlinear characteristics of materials, the lubrication state of bearings, and system noise interference. These factors have not been taken into account in this study.

## Figures and Tables

**Figure 1 sensors-25-01691-f001:**
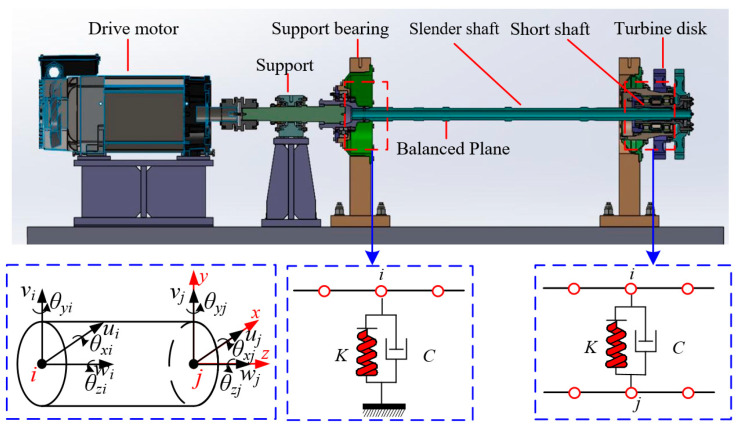
Aero-engine turbine rotor system structure.

**Figure 2 sensors-25-01691-f002:**
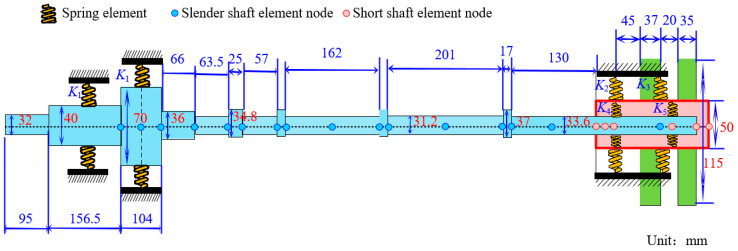
Rotor system dynamics model.

**Figure 3 sensors-25-01691-f003:**
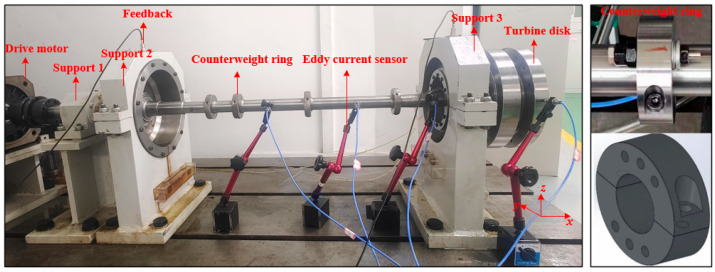
Turbine rotor system test bench.

**Figure 4 sensors-25-01691-f004:**
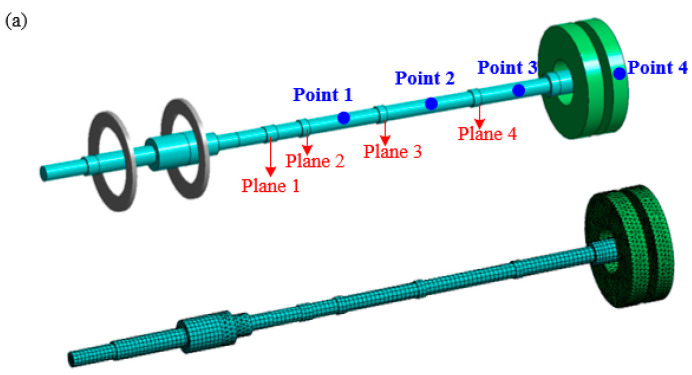
Turbine rotor system: (**a**) finite element model and (**b**) vibration mode shapes.

**Figure 5 sensors-25-01691-f005:**
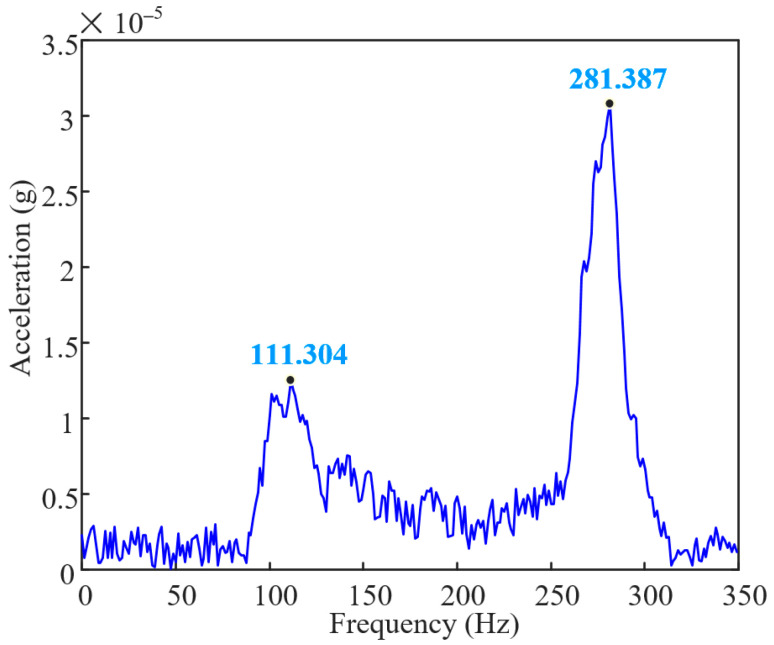
Frequency response function of turbine rotor system.

**Figure 6 sensors-25-01691-f006:**
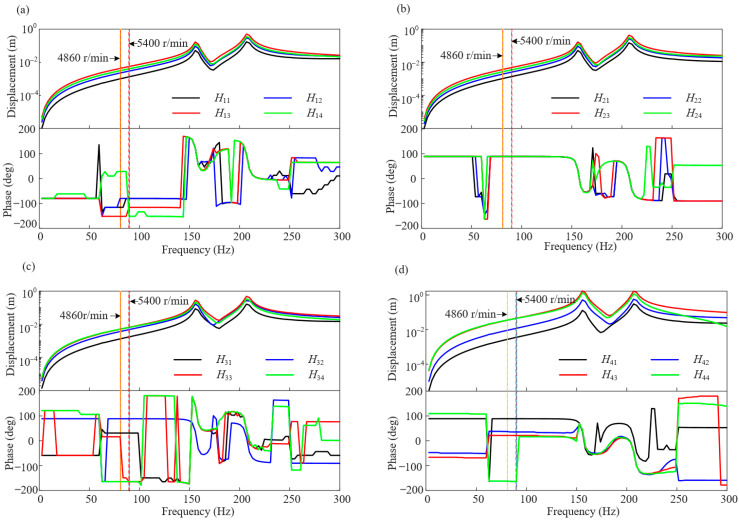
Frequency response curve diagram at 5400 r/min: (**a**) point 1, (**b**) point 2, (**c**) point 3, and (**d**) point 4.

**Figure 7 sensors-25-01691-f007:**
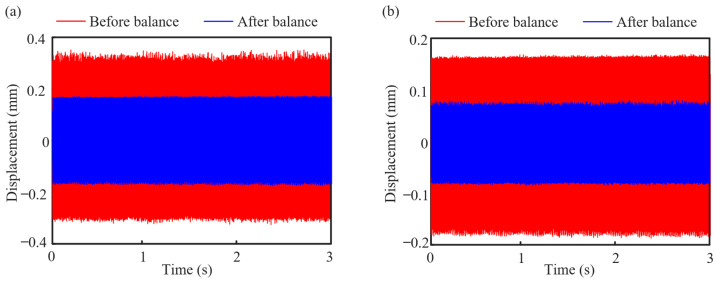
Comparison of vibration response before and after dynamic balancing at 5400 r/min: (**a**) point 1 and (**b**) point 3.

**Figure 8 sensors-25-01691-f008:**
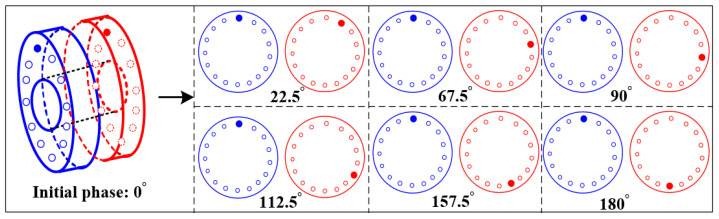
Schematic diagram of unbalanced phase difference combination.

**Figure 9 sensors-25-01691-f009:**
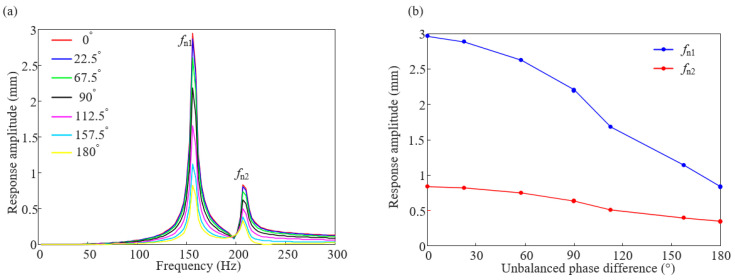
Harmonic response amplitude diagram: (**a**) vibration response curve and (**b**) vibration amplitude change curve.

**Figure 10 sensors-25-01691-f010:**
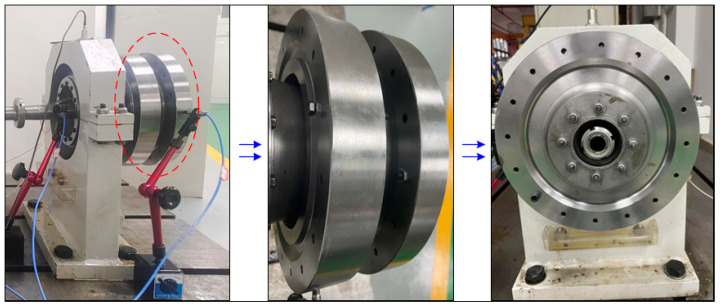
Turbine disk phase difference combination.

**Figure 11 sensors-25-01691-f011:**
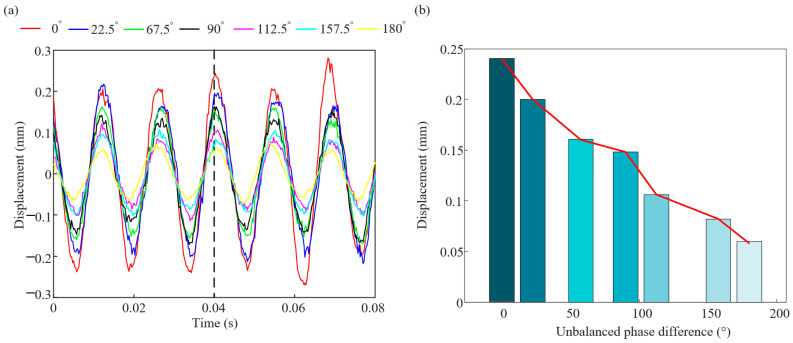
The measured rotor vibration response under 7 kinds of phase difference combination: (**a**) vibration response and (**b**) vibration amplitude change bar.

**Figure 12 sensors-25-01691-f012:**
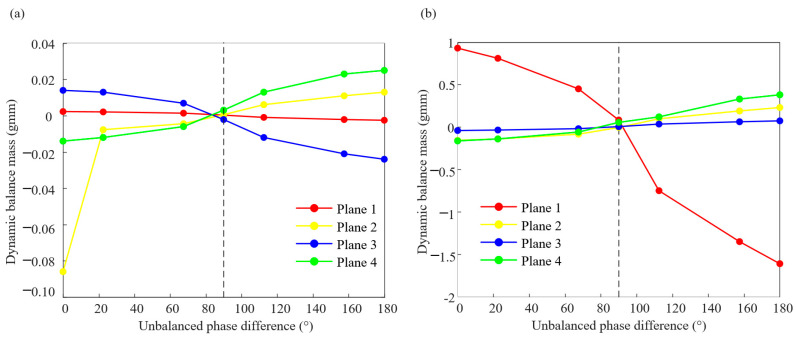
Balancing counterweight under 7 kinds of phase difference combination: (**a**) 4860 r/min and (**b**) 5400 r/min.

**Table 1 sensors-25-01691-t001:** Rotor structure and material parameters.

Parameter	Value
Total mass of disk (kg)	14.58
Disk diameter moment of inertia (kg·m^2^)	0.12
Disk polar moment of inertia (kg·m^2^)	0.13
density (kg/m^3^)	7800
Modulus of elasticity (Pa)	2.6 × 10^11^
Poisson’s ratio	0.3
*K*_1_ (N/m)	1 × 10^8^
*K*_2_ (N/m)	1 × 10^7^
*K*_3_ (N/m)	1 × 10^7^
*K*_4_ (N/m)	2 × 10^8^
*K*_5_ (N/m)	1 × 10^7^

**Table 2 sensors-25-01691-t002:** Balancing counterweight schemes.

Balancing Speed (r/min)	Counterweight Plane 1	Counterweight Plane 2	Counterweight Plane 3	Counterweight Plane 4
4860	0.0023 gmm, −30.91°	−0.086 gmm, −88.31°	0.014 gmm, 61.62°	−0.014 gmm, 58.13°
5400	0.93 gmm, −53.87°	−0.157 gmm, −89.99°	−0.041 gmm, 99.56°	−0.163 gmm, −16.64°

**Table 3 sensors-25-01691-t003:** Balancing counterweight schemes at 4860 r/min under 7 kinds of phase difference combination.

Balancing Speed (r/min)	Counterweight Plane 1	Counterweight Plane 2	Counterweight Plane 3	Counterweight Plane 4
0°	0.0023 gmm, 30.9°	−0.086 gmm, −88.31°	0.014 gmm, −61.62°	−0.014 gmm, 58.13°
22.5°	0.0021 gmm, 29.2°	−0.0077 gmm, 102.61°	0.013 gmm, −96.62°	−0.012 gmm, −111.51°
67.5°	0.0014 gmm, −47.11°	−0.0044 gmm, 97.79°	0.0069 gmm, −32.3°	−0.006 gmm, −110.7°
90°	0.00033 gmm, −152.6°	0.00058 gmm, 92.12°	−0.0021 gmm, 61.84°	0.0031 gmm, −110.97°
112.5°	−0.00092 gmm, −50.09°	0.0061 gmm, −7.2°	−0.012 gmm, 96.64°	0.013 gmm, 67.13°
157.5°	−0.0021 gmm, 3.03°	0.011 gmm, −3.08°	−0.021 gmm, 37.36°	0.023 gmm, 46.21°
180°	−0.0025 gmm, 18.79°	0.013 gmm, −4.06°	−0.024 gmm, −24.77°	0.025 gmm, 59.16°

**Table 4 sensors-25-01691-t004:** Balancing counterweight schemes at 5400 r/min under 7 kinds of phase difference combination.

Balancing Speed (r/min)	Counterweight Plane 1	Counterweight Plane 2	Counterweight Plane 3	Counterweight Plane 4
0°	0.93 gmm, −53.87°	−0.157 gmm, −89.99°	−0.041 gmm, 99.56°	−0.163 gmm, −16.64°
22.5°	0.81 gmm, −77.59°	−0.14 gmm, −140.86°	−0.035 gmm, 142.62°	−0.14 gmm, −0.4°
67.5°	0.45 gmm, −101.51°	−0.084 gmm, −174.84°	−0.019 gmm, −169.6°	−0.059 gmm, −24.99°
90°	−0.083 gmm, −118.86°	−0.0043 gmm, 122.2°	0.0048 gmm, −167.84°	0.055 gmm, 55.3°
112.5°	−0.75 gmm, 85.6°	0.097 gmm, 117.39°	0.035 gmm, −26.03°	0.12 gmm, −69.85°
157.5°	−1.35 gmm, 57.79°	0.19 gmm, −94.84°	0.062 gmm, −159.35°	0.33 gmm, −79.6°
180°	−1.61 gmm, 18.79°	0.23 gmm, −161.2°	0.073 gmm, −119.99°	0.38 gmm, −40.4°

## Data Availability

Data are contained within the article.

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
