# Peer review of "Study on the Influence of Unbalanced Phase Difference Combinations on Vibration Characteristics of Rotor Systems"

_sensors, 2025, doi:10.3390/s25061691_

Round 1

Reviewer 1 Report

Comments and Suggestions for Authors

 In this paper, the cantilever rotor dynamic model is established, and the rotor system under unbalanced excitation is balanced by the transfer function method, which effectively reduces the vibration of the unbalanced rotor. The experimental test and simulation analysis of rotor vibration response under different unbalanced phase difference combinations are carried out, and the influence of unbalanced phase difference combinations on unbalance and dynamic balance is analyzed. The overall logical structure of the paper is clear, and it has certain academic value and application potential. However, there are still some areas for improvement in some details, and it is recommended that the author make revisions and improvements.

  1. The author mentions that existing studies have paid little attention to the effect of unbalanced phase combination on the vibration characteristics of rotating subsystems, but the literatures cited in the paper (e.g. Wang et al. [29], Chen et al. [31]) have involved the analysis of the effect of phase difference. Can you more clearly explain the innovation of this paper compared to existing research (especially recent literature)?

  1. In dynamic modeling, the hollow slender shaft is equivalent to a solid shaft, and the turbine disc is simplified into a concentrated mass disc. Do these simplifications lead to distortions in key dynamical properties such as gyroscopic effects or higher-order modes?

  1. In model verification, the first-order and second-order natural frequency errors of simulation and experiment are 6.2% and 7.3%, respectively. Are sources of error (such as boundary condition simplification, material parameter assumptions, sensor accuracy, etc.) discussed? Will such errors affect the reliability of subsequent vibration response analysis?

  1. Seven phase difference combinations ranging from 0° to 180° were selected, but the theoretical basis for selecting these specific angles (such as uniform distribution or optimization based on modal sensitivity) was not stated. Has the representativeness of these combinations been verified by pre-experiment or theoretical analysis?

  1. It is pointed out that the vibration amplitude of the inverse phase combination is significantly reduced, and the equilibrium mass is minimum at 90° phase difference. Does this conclusion apply only to cantilever rotor systems? Are there any operating conditions limitations (such as supercritical speed or transient processes) in practical engineering applications? The generalizability and limitations of the conclusions should be discussed.

  1. Some of the references were published in 2024, but recent studies (e.g. 2023-2024) in some key areas (such as the effect of phase difference on multi-disk rotor systems) are less cited. Are recent advances in related fields (such as machine learning-based phase optimization methods) missing?

  1. Figure 11 shows that the experimental and simulation trends are consistent, but does not provide quantitative error analysis (such as root-mean-square error or correlation coefficient). Can the reliability of the results be enhanced by statistical methods such as error band analysis? Are the experimental data reproducible (e.g. multiple measurements averaging)?

Author Response

Please refer to the attachment for specific reply.

Reviewer 2 Report

Comments and Suggestions for Authors
  1. The description of the rotor system lacks rigor. In Fig. 1, it seems that there is only one supporting bearing on the left side of the rotor system. However, there are actually two supporting bearings, as shown in Fig. 2.
  2. The rotor system contains two turbine disks, but only one mass is provided in Tab. 1. Are the masses of the two turbine disks the same?
  3. Fig. 4(b), the shape of the second-order mode appears to be U-shaped, rather than S-shaped.
  4. Fig. 6, the markings of 5400 rpm and 4860 rpm seem incorrect.
  5. What position's response is shown in the Fig. 9?
  6. In the dynamic balance analysis of the rotor, the counterweight is a vector, which includes two variable quantities: amplitude and phase. In this study, counterweights are added to two turbine discs respectively. Even if only the phase difference is changed, the equivalent amplitude of the two counterweights also changes. Therefore, the obtained results are difficult to be summarized as merely the influence of phase difference combination. Further analysis and discussion are required to enhance the understanding of the subject matter.

Author Response

(The authors gave the same response as above.)

Round 2

Reviewer 2 Report

Comments and Suggestions for Authors

None